



# An appraisal of the value of simulated weather data for quantifying coastal flood hazard in the Netherlands

Cees de Valk[1, *] and Henk van den Brink[1, *]

[1]KNMI, PO Box 201, 3730 AE De Bilt, The Netherlands

**Correspondence:** Cees de Valk (cees.de.valk@knmi.nl)

**Abstract.** With recently updated safety norms, assessment of flood safety in the Netherlands requires return values for coastal sea level and surface stress for a wide range of return periods up to $10^7$ years. Estimates from measurements are highly uncertain. To reduce the uncertainty, a possible solution could be to replace measurements by simulated weather datasets much larger than a typical measurement record. However, systematic errors in simulations can easily outweigh any gains in precision.

Combining insights from physics and extreme value theory with evidence from data, we argue that even as stress from present-day weather prediction models may be too high or too low, these data are suitable for estimating the shape of the upper tail of the distribution function of stress, and that this extends to simulated data of water level along the Dutch coast. As scale and location parameters can be estimated with sufficient precision from relatively short measurement records, we estimate return values from a combination of measurements (for scale/location of water level) and simulated data (for shape), assess their

uncertainty, and discuss strengths and limitations of the approach and prospects for further exploiting simulated weather data to assess flood risk.

## 1 Glossary

*High water, abbrev. HW*: the maximum height reached during a rising tide (including the meteorologically forced contribution) relative to a fixed datum (Hicks, 1989) (also hoogwaterstand (Dutch), Hochwasserstand (German)). *Skew surge*: the difference

between the high water and the astronomical high water predicted for the same cycle. *Return value*: the value exceeded with a frequency of $1/T$, with $T$ the return period in years.

## 2 Introduction

Flooding due to high river discharge and/or storm surge is an existential threat to many low-lying river deltas (e.g. Tessler et al, 2015; Ward et al, 2018). Most of The Netherlands is formed as a delta of Rhine, Meuse and Scheldt on the shallow North

Sea, which renders the country particularly vulnerable to floods due to storm surge (Gerritsen, 2005). The coast is protected by dunes, sea dikes and storm surge barriers. By law, these are required to meet local safety norms which specify the maximal allowed frequency $\mu = 1/T$ of failure, with $T$ the return period (MinBZK, 2023). These norms are based on the consequences



of a local failure of the flood protection (Kok et al., 2018). The norm for the return period of failure may be as high as $10^6$ years (in the case of a stretch of dike near a nuclear power plant).

To check if the flood protection satisfies the local safety norm, a range of models is available (see https://iplo.nl/thema/water/ applicaties-modellen/software-beoordelings-ontwerpinstrumentarium/). One of these is HYDRA-NL, a probabilistic model of the hydraulic loads (including their uncertainties) for the assessment of dikes. It provides for example return values of water level and surface wave height and period in front of the dike, and return values of wave overtopping discharge. For the coast, these are computed from wind direction-dependent return values of water level and wind speed and the dependence between

wind and water level, interpolating from a database of input and output of nearshore wave computations using the SWAN model (Booij et al, 1999). In these computations, the wind is taken as spatially uniform and either constant or evolving according to a fixed pattern determined from data of historical storms, and offshore wave boundary conditions are determined from wind speed and direction using relationships derived from offshore wave and wind measurements (Groeneweg et al, 2010). For some areas, simulated currents with different tidal phases relative to the wind speed peak are included in the nearshore wave

simulations as well.

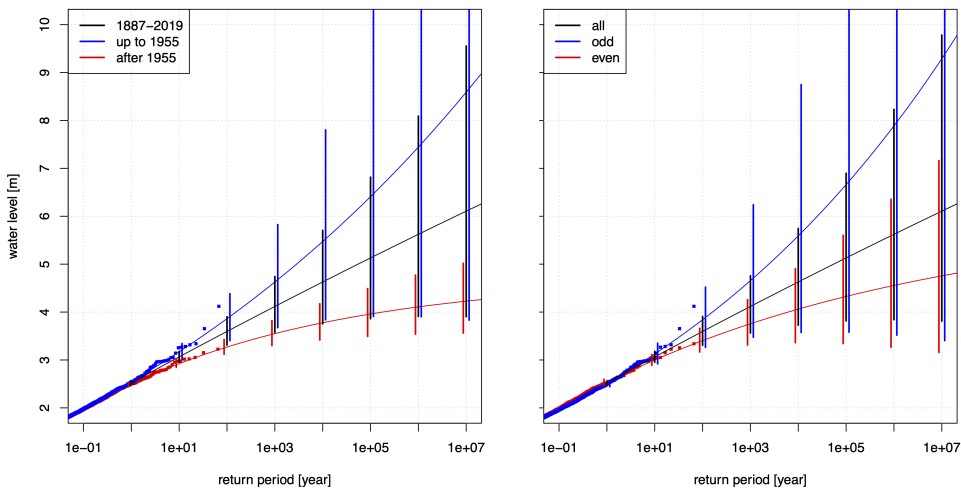

**Figure 1.** Estimates of return values of sea level at Hoek van Holland from trend-corrected measurement data (black curve) with 95% confidence intervals (thick black lines). Left: data and estimates from measurements made up to (blue) and after (red) 1955. Right: data and estimates from even (red) and odd (blue) years. Tail: Generalized Pareto (see Section 4).

Return values of coastal high water and wind speed are usually estimated from measurement or reanalysis data, generally covering less than 150 years (e.g. Woodworth et al, 2016; Ramon et al, 2018). For long return periods, such estimates are highly uncertain (e.g. Dillingh et al, 1993; Caires, 2009). As an illustration, Fig. 1 shows estimates of return values of water level at Hoek van Holland (tidal station 2 in Fig. 2) derived from the 1% highest trend-corrected high water values measured since 1887 (black curve) with their 95% confidence intervals (vertical bars); see Section S1 in the Supplementary Materials

for further details and discussion. For the longest return periods, the confidence intervals are very wide: about 2 m for a return



period of 10,000 years, and 6 m for a return period of $10^7$ years. The estimates in the landmark study Dillingh et al (1993) have even wider confidence intervals. Also shown in Fig. 1 (left) are the estimates from the measurements up to 1955 (blue) and after 1955 (red) along with the corresponding data points. These estimates are very different, spanning almost the entire widths

of the confidence intervals of the estimates from the complete dataset. Further, splitting the data of odd and even years (Jul to Jun), similarly large differences are found; see Fig. 1 (right). This underscores that the large uncertainty implied by these confidence intervals is realistic (see Supplement S1.2 in the Supplementary Materials for further discussion).

In fact, high uncertainty of return value estimates also prevents to properly account for this uncertainty (e.g. Ditlevsen and Madsen, 1996, Section 3.4) in flood risk estimation: this requires that the uncertainty can be reliably quantified, but that

need not be the case. For example, in Figure 1 (left), the estimates from the complete dataset (presumably the most accurate) lie far outside the confidence intervals derived from the data after 1955, so if only the latter were available, the uncertainty would be seriously underestimated. A further consequence of high uncertainty is that flood-protecting structures need to be over-designed, which is costly.

Hence, there is ample reason to look for ways to reduce the uncertainty in return values of coastal high water and wind

stress. Several approaches to reduce uncertainty have been proposed. One is refinement of the extreme value analysis, briefly addressed in Section 4. Other approaches focus on increasing the amount of data available, e.g. by extracting useful information from sparse/non-systematic records of historical floods, by combining data from different sites, or by making use of simulated weather data.

The potential of using non-systematic historical records of floods (e.g. Reis Jr and Stedinger, 2021; Parkes and Demeritt,

2016; Hamdi et al, 2015; Van Gelder, 1996; Baart, 2015) is determined by the availability and quality of such data. For coastal floods in the Netherlands, it can at best increase the effective size of the dataset by a factor $< 4$ (Van Gelder, 1996; Baart, 2015), hence reduce the sampling error (the error due to the limited size of the dataset) by a factor $< 2$, but this is not likely to be achieved with data of only a few events. Further limitations are the large uncertainty in reconstructed values of historical HW or skew surge (Baart, 2015) and strong assumptions which seem hard to justify and to check; for example, the annual

mean high-tide level is largely unknown before measurements were systematically recorded, hence any proposed correction for its change over time is highly uncertain (see e.g. Fig. S3 in the Supplementary Materials).

Pooling of data from different sites, either by traditional regional frequency analysis (e.g. Bardet, 2011) or by estimation of a spatially varying distribution of the annual maxima, assuming that it has a uniform shape and smoothly varying location and scale parameters (Calafat and Marcos, 2020). Spatial dependence of storm surge is considerable (Calafat and Marcos,

2020), which reduces the potential gains from pooling. Calafat and Marcos (2020) report a reduction of the sampling error in estimated 50-year return values by a factor 2.

In contrast, the use of simulated data for the statistical analysis of extremes (e.g. van den Brink et al, 2004, 2005; van den Brink and Können, 2008, 2011; van den Brink and de Goederen, 2017; Kelder et al., 2020, 2022a, b) could be a game changer: in principle, the amount of data is unlimited, and very large datasets of simulated weather representative of the climate of recent

decades arse already available. An example is the SEAS5 ensemble seasonal (re)forecast archive from ECMWF (ECMWF, 2018a, b). For a single site, SEAS5 offers about 8000 years of simulated near-surface wind and shear stress and mean sea level



pressure (mslp) which can be used to simulate nearshore wave conditions and coastal water levels; see Section 3. Use of these data as replacement of measurement records covering up to 150 years can reduce the sampling error in return values by a factor $\sqrt{8000/150} > 7$. Furthermore, model-simulated stress can be used directly for forcing of hydrodynamic models, avoiding the precarious step of deriving stress from wind (e.g. Van Nieuwkoop et al, 2015).

However, are simulated data good enough for estimation of the return values of coastal sea level and wind stress for nearshore wave modelling? This is the question addressed in the present paper, focusing on the Dutch coast.

Unlike (re)analyses of the weather of the past (e.g. Bauer et al, 2015; Hersbach et al., 2020), simulated weather data are not tightly constrained by weather observations. Models simulating weather do nothing more than solving the physical equations that govern the behaviour of the atmosphere. Starting from an atmosphere at rest, switching on the sun and rotating the earth at its known speed will soon result in realistic chaotic behaviour (Lorenz, 1969; Zhang et al, 2019): differential heating of the poles with respect to the equator results in a temperature gradient driving the motion of air masses which (due to rotation) forms jet streams, on which (due to barotropic and baroclinic instability) extratropical cyclones will develop. However, the devil is in the details: is the temperature gradient between the equator and the poles large enough? Is the vertical structure of the temperature difference realistic, and consequently: is the jet stream not too strong or too weak, and climatologically at the right latitude? And is the number of cyclones correct, as well as their inter-arrival times?

For the present application, there are reasons to expect that model-generated weather data may be useful. Models have gradually improved over the last decades, due to the increase in resolution and the improvement of parameterizations of subgrid processes (Bauer et al, 2015). The large spatial and temporal scales of extratropical storms and storm surges on the North Sea make simulations fairly insensitive to model resolution. Only for shallow estuaries, stress variations on smaller scales may be relevant. Land surface roughness is not really an issue, as severe loads on coastal structures tend to coincide with wind from sea. Even the parameterisation of the drag by the sea surface is of secondary importance, as it is not the wind that drives surges and waves, but the stress in the boundary layer (Zweers et al, 2012). The same holds for the mean inter-arrival time of storms, as we are mainly interested in very extreme surges (rarer than 1:100 years), which can safely be regarded as independent.

In the present study, we perform a more targeted assessment of the value of simulated data for the estimation of return values of high water and stress. Starting point is that most of the uncertainty in return values for long return return periods is associated with uncertainty in the shape of the tail of the distribution function; scale and location parameters can be estimated with sufficient precision from relatively short measurement records (Section 4). We argue in Section 5 that even as wind stress from present-day weather prediction models may be too high or too low, the SEAS5 data are suitable for estimating the shape of the tail of stress over the North Sea, and we argue in Section 6 that this extends to coastal water levels simulated from the SEAS5 stress and mean sea level pressure (mslp) fields. Based on these findings, return values of stress and water level are estimated using SEAS5 stress and coastal water level data simulated from SEAS5 data for the estimation of tail shapes, and the uncertainties are assessed (Section 7). In addition, we examine the value of SEAS5 data and simulated water level data for estimating dependencies between variables in the extreme range. Strengths and weaknesses of the proposed approach are evaluated, and the prospects for future improvements and extensions are indicated (Section 8).

Labels of sections, figures, tables and equations containing "S" refer to the Supplementary Materials.



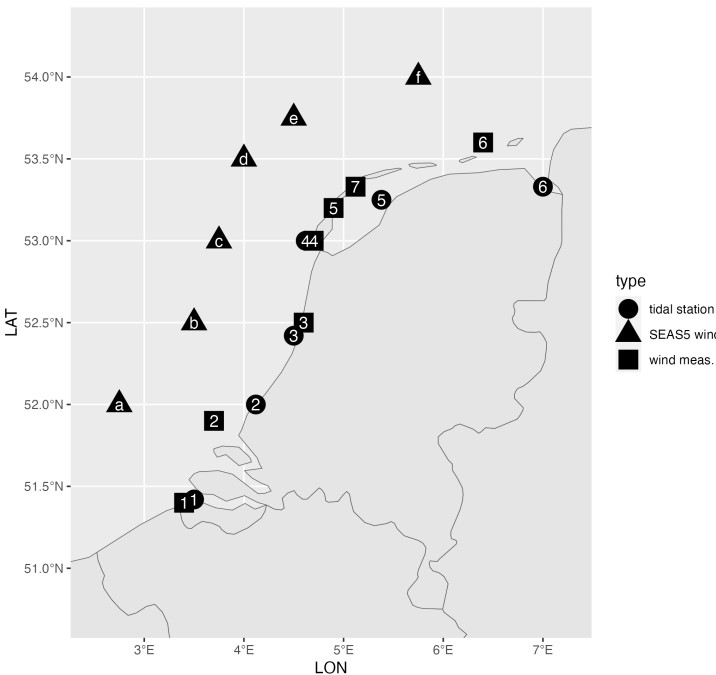

**Figure 2.** Locations of tidal stations (o), SEAS5 output grid points (△), and weather stations (□). Numbering refers to main text.

## 3 Data

*SEAS5 stress and mslp.* The primary source of simulated weather data for the present study is the archive of 6-hourly SEAS5 ensemble seasonal (re)forecasts from the IFS cycle 43r1 model of ECMWF (ECMWF, 2018a, b) with nominally 35 km resolution. An ensemble of 25-50 seasonal forecasts over 7 months was started by ECMWF at the beginning of each month from 1981 onwards. We selected data of near-surface shear stress (vector), near-surface wind $u_{10}$ (vector) and mslp, discarding the first month of each run to ensure that the extremes of stress from different ensemble members are independent: this is verified in Section S2. Stress and mslp were used to force tide-surge predictions (see below) and stress at the 6 points shown by triangles in Fig. 2) was selected for statistical analysis. These points were chosen at roughly 50 km from the shore to avoid any influence of land on surface roughness.

*Stress and mslp from climate models and reanalyses.* Data from climate model simulations used for checking of assumptions include a 16-member ensemble of EC-Earth3 runs for the present-day climate, and dynamic downscaling of these data using the RACMOv2.3 regional climate model (van Dorland et al, 2023). Furthermore, we used data from two reanalyses: ERA5, the most recent global reanalysis by ECMWF covering the years 1979-2019 (Hersbach et al., 2020), and KNW, a regional downscaling of ERA-Interim (Dee et al, 2011) (the predecessor of ERA5) for the North Sea over 1979-2019, produced by KNMI using the mesoscale model HARMONIE (Bengtsson et al., 2017). Validation of reanalysis wind against measure-





ments is presented in Belmonte Rivas and Stoffelen (2019); Kalverla et al. (2019); Tetzner et al (2019); Wijnant et al (2019) for ERA5 and in Stepek et al (2015); Wijnant et al (2015, 2019) for KNW; see also further references in this articles. In particular, the empirical tail distributions of KNW wind speeds and of mast measurements are found to agree well (Stepek et al, 2015). These studies support the use of these two reanalyses for validation of statistics of stress from SEAS5.

*Wind measurements.* Measurements of hourly mean wind from seven KNMI weather stations were used to check the statistics of stress from SEAS5 data: along the coast from SW to NE, these are Cadzand (1), LEG (Lichteiland Goeree) (2), IJmuiden (3), Texelhors (4), Vlieland (5), West Terschelling (7) and Huibertgat (6), indicated by squares in Fig. 2.

*Coastal tide gauge measurements.* Time-series of HW relative to the national NAP-datum derived from measurements were provided by Rijkswaterstaat. Here, we consider (from SW to NE) data from Vlissingen (1), Hoek van Holland (2), IJmuiden (3), Den Helder (4), Harlingen (5) and Delfzijl (6), indicated by disks in Fig. 2. Records begin around 1880-1890 (1, 2, 3, 6) and around 1932 (4, 5). They were corrected for a long-term trend using a smooth curve fitted to the record of annual mean HW; see Section S3. The corrected values are representative for the year 2019.

*Coastal water levels simulated from SEAS5 data.* We simulated still water levels on the North Sea using the WAQUA DCSMv5 shelf sea tide and surge prediction model with 8 km resolution (Ridder et al, 2018), forced by SEAS5 stress and mslp, further referred to as SEAS5/DCSMv5 data (van den Brink, 2020). This resulted in 10-min time series for over 300 locations along the Dutch coast, including the 6 tidal stations listed above, from which HW was derived using astronomical tide computed using DCSMv5 without variation in meteorological forcing. Also, a simulation was performed with the stress forcing increased by 10%.

## 4  Tail approximation and estimation

For approximation of a tail beyond the range of observations, its local shape needs to become stable in some sense as the probability of exceedance tends to zero. We call this a regular tail. Non-regularity may occur for example in mixtures of distributions of subpopulations with different tails (e.g. frontal vs. convective subdaily rainfall intensity) or due to a change in the physics of a process near some threshold (e.g. the limitation of sea surface wave growth by breaking on shallow water). Different types of regularity assumptions may be made, of which we discuss two here in the context of smooth tails. For a distribution function $F$ with a smooth nonzero density, let $q$ be the function satisfying

$$1 - F(q(y)) = e^{-y}, \qquad y \geq 0, \tag{1}$$

so $q$ (the "quantile function") expresses a quantile of $F$ in terms of the quantile of the exponential distribution for the same probability. The classical regularity assumption employed in extreme value theory can be formulated in terms of the dimensionless function $\tilde{\gamma}$ defined by $\tilde{\gamma}(y) := q''(y)/q'(y) = (\log q'(y))'$ (with $'$ indicating the derivative) as

$$\lim_{y \to \infty} \tilde{\gamma}(y) = \lim_{y \to \infty} (\log q'(y))' = \gamma \in \mathbb{R} \tag{2}$$



(e.g. de Haan and Ferreira, 2006, Corollary 1.1.10). By integration, this implies

$$\lim_{y \to \infty} \frac{q(y+x) - q(y)}{q'(y)} = \frac{e^{x\gamma} - 1}{\gamma}, \quad x \in \mathbb{R} \tag{3}$$

(to be read as $x$ if $\gamma = 0$). Eq. (3) provides a recipe for approximating a higher quantile $q(y+x)$ exceeded with probability $e^{-(y+x)}$ from a lower quantile $q(y)$ exceeded with probability $e^{-y}$. It is one of the formulations of the Generalized Pareto (GP) tail limit employed in classical peaks-over-threshold analysis (Leadbetter, 1991). The GP tail limit (3) is equivalent to the classical assumption that the distribution of the scaled and shifted maximum of a random sample $X_1, ..., X_n$ of size $n$ tends to a nonconstant limit (de Haan and Ferreira, 2006), the GEV (generalized extreme value) distribution. In particular for large datasets, estimation of $\gamma$ based on GEV fits to annual maxima is convenient.

There is ample evidence that $\gamma = 0$ is appropriate for (powers of) wind speed (Cook, 1982; Harris, 2005; van den Brink and Können, 2008), and therefore also for stress, surge and HW (in the absence of large-scale flooding of land); note that a power of a positive random variable with $\gamma = 0$ still has $\gamma = 0$. Another regularity assumption, particularly useful as refinement of (2) if $\gamma = 0$, replaces the derivatives to $y$ in (2) by derivatives to $\log y$:

$$\lim_{y \to \infty} y \big( \log(y q'(y)) \big)' = \lim_{y \to \infty} y \tilde{\gamma}(y) + 1 = \beta \in \mathbb{R}. \tag{4}$$

This leads to the Generalized Weibull (GW) limit (de Valk, 2016a)

$$\lim_{y \to \infty} \frac{q(y\lambda) - q(y)}{y q'(y)} = \frac{\lambda^\beta - 1}{\beta}, \quad \lambda > 0. \tag{5}$$

In case $\beta > 0$ (de Valk, 2016a; Gardes and Girard, 2015), tail approximations based on (5) take the form of a 3-parameter (translated) Weibull distribution, i.e. $\log P(X > x) = -((x-b)/a)^{1/\beta}$ for some $b \in \mathbb{R}$ and $a > 0$. This helps in interpreting the GW tail shape parameter $\beta$: e.g. 1 gives an exponential tail, 0.5 a Rayleigh tail, etc. Unlike (3), (5) approximates $q$ over a probability range which increases rapidly with $y$, which is potentially useful for estimating return values for very long return periods.

Many estimators of $\gamma$ from the $k$ highest values in a sample of size $n$ can be regarded as estimators of a smoothed $\tilde{\gamma}(y)$ at $y$ not far from $\log(n/k)$ (e.g. Beirlant et al, 1996). Similarly, estimators for $\beta$ such as de Valk and Cai (2018) can be regarded as estimators of a smoothed $y\tilde{\gamma}(y) + 1$ at $y \approx \log(n/k)$. In the next sections, we sometimes analyse tail shape differences using estimates of $\gamma$ from annual maxima (to compare values of $\tilde{\gamma}(y)$ from different data sources at a fixed $y$), and in other instances show estimates of GW shape parameter $\beta$ from the upper $k$ values as a function of sample fraction $k/n$, to highlight the dependence of $y\tilde{\gamma}(y) + 1$ on $y \approx \log(n/k)$. It is important to keep in mind that these show different, but essentially equivalent representations of the local tail shape. In these comparisons, they are used only as diagnostics, whose meaning does not depend on the validity of a regularity assumption like (2) or (4).

Because we want to focus on the choice of the data source, we refer to Appendices F and G for the technical aspects of tail estimation. In brief, the selection of methods and settings was informed by the results of a preparatory study (de Valk and van den Brink, 2023a), employing Monte Carlo simulation based on distribution functions for wind speed, skew surge and HW estimated from the long archive of SEAS5/DCSMv5 data using a refined method, which display plausible deviations from





limiting GP and GW tail shapes. In these simulations, the uncertainty of return values for long return periods is dominated by
the uncertainty in the shape estimates, in accordance with theory (e.g. de Haan and Ferreira, 2006; de Valk and Cai, 2018).
In particular, they show that scale and location (e.g. $yq'(y)$ and $q(y)$ in (5)) can be estimated with sufficient accuracy from
relatively short records similar in size to typical measurement records, if the shape is estimated from a record of the size of the
local SEAS5 data. Estimates of return values for $10^7$ year for skew surge and HW appear te be of similar accuracy, so there
is nothing to be gained from estimating return values of HW by convolution of estimated distributions of skew surge and tide.
Furthermore, if the tail shape of HW is estimated from a record similar in size to a measurement record, the GW tail produces
considerably more accurate estimates than the GP tail, but this difference largely disappears if a record of the size of the local
SEAS5 data is used for shape estimation. However, for wind speed and for pseudo-wind speed derived from SEAS5 stress
using a Charnock relation, the GW tail performs considerably better than the GP tail, also when very long records are used for
shape estimation (de Valk and van den Brink, 2023a, Appendix C). Based on these findings, the GW tail was selected for the
estimation of return values in Section 7.

## 5    Can we trust the tail shape of stress over the North Sea derived from simulated data?

In this section, we argue that (1) large-scale metrics of storm intensity such as mslp have regular tails, (2) climate models of
sufficient resolution show high skill at simulating extratropical cyclones, and agree about the tail shapes of mslp distributions,
(3) stress in the boundary layer is closely linked with overall storm intensity, and (4) the tail of the stress distribution is also
regular, and its shape can be estimated reliably from climate model simulations. We will use the term climate model for any
model run for a considerable time without data-assimilation, which includes the selected SEAS data used in this study (Section
3).

### 5.1    Large-scale metrics of storm intensity are expected to have regular tails

Coastal flood risk in the Netherlands is determined by extra-tropical cyclones in the winter storm season. Extratropical cy-
clones are a well-defined phenomenon; variations in their development constitute a continuum (Graf et al, 2017; Seiler, 2019).
Therefore, tail anomalies due to mixing subpopulations with distinctly different tails are not an issue. Furthermore, the growth
of extra-tropical cyclones does not appear to involve sharp transitions or limits at a certain intensity level which could give rise
to a tail anomaly.

Common metrics of overall storm intensity are for example the spatial-temporal minimum of mslp or maximum of relative
vorticity at 850 hPa. Because the density of storm tracks varies smoothly, we may expect that the values at a fixed location
also have smooth tails. For illustrations, see for example Fig. 2 in Sterl (2009), or Fig. 3, showing the empirical return values
of daily-averaged mslp from 147 years of measurements at Thyboron (8.2E, 56.7N) on the West coast of Denmark, from the
RACMOv2.3 and EC-Earth3bis climate model simulations (see Section 3), and from SEAS5. A low mslp at this site indicates
the presence of a pressure low nearby, hence NW wind over the North Sea and positive surge at the Dutch coast. The modelled



tails are indeed smooth (very long series); the measurement record is much shorter, hence the empirical distribution of the
annual minima is more noisy, but agrees reasonably well with the modelled tails.

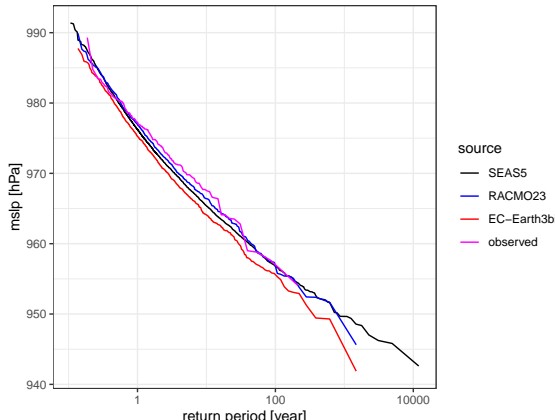

**Figure 3.** Empirical return values from annual maxima of daily mean mslp near 8.2E, 56.7N from SEAS5 (black), RACMOv2.3 (blue) and
EC-Earth3bis (red) and from measurements at Thyboron (DK) over 1874-2020 (magenta).

## 5.2  Climate models of sufficient resolution show high skill at simulating extratropical cyclones and agree about the tail shapes of mslp distributions

More than a decade ago, climate models already produced skilful simulations of extra-tropical cyclones when comparing the
intensity, tracks and storm structure to reanalysis data (Catto et al, 2010; Bengtsson et al, 2009; Jung et al, 2012). Then and
now, a sufficiently high resolution (25-50 km) is a requirement (Priestley and Catto, 2022), but further increase in resolution
offers limited benefit (Jung et al, 2012). Remaining limitations appear to concern mainly the flow in the upper atmosphere
(250 hPa). The resolution of SEAS5 of about of $0.25°$ should thus be sufficient for the simulation of coastal surges and waves
requiring a long fetch and duration to develop. Indeed, van den Brink (2020) shows that the limited resolution of SEAS5 only
gives a 1.5% reduction in surge and a further 3.5% due to 6-hourly sampling of the SEAS5 output, in comparison with hourly
sampled 2.5 km resolution HARMONIE model output (Section 3).

The high skill is particularly evident in the tail shapes of storm intensity metrics. In Fig. 3, the shapes and scales of dis-
tributions of annual minima of daily mean mslp from RACMOv2.3, EC-Earth3bis and SEAS5 show remarkable agreement:
the main differences are small (but significant) shifts in pressure, and even smaller differences in scale. All agree closely with
the empirical tail from the observations and are capable of generating deeper depressions than observed at this location. Fig. 4
displays estimates of the GEV parameters from the annual maxima of daily-averaged mslp over the North Sea from effectively
1040 years of EC-Earth3bis and its downscaling with the regional RACMOv2.3, and their difference. It shows excellent spatial
agreement between these models. The standard error of the estimates of the shape parameter $\gamma$ is about 0.02, implying that the
cluttered difference in shape is insignificant at all grid points. This confirms that the considerable differences in resolution and





boundary layer parameterisation of these models leave the shape of the mslp tail virtually unaffected. This is remarkable, as these simulated storms and their climatologies are generated completely by the models, unlike numerical weather forecasts.

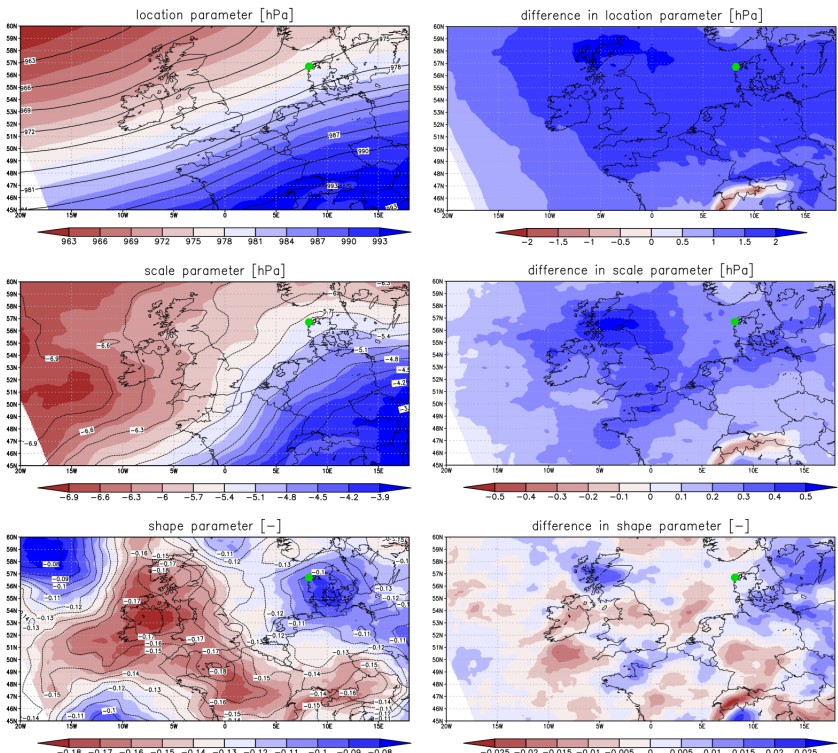

**Figure 4.** Spatial patterns of estimates of resp. location, scale and shape γ (see Section 4) of the GEV distribution fitted to the annual minimum mslp. Left: from RACMOv2.3 (colours) and from EC-Earth3bis (contours). Right: differences (RACMOv2.3 minus EC-Earth3bis) The green dot indicates Thyboron.

### 5.3  Stress in the boundary layer is closely linked with overall storm intensity

Stress in the boundary layer over sea depends on the wind aloft, stability, and the interaction with the sea surface, which are all different in different sectors of the storm. In turn, the stress affects the evolution of the storm by barotropic (e.g. Ekman
pumping) and baroclinic potential vorticity generation in the boundary layer. In a modelling study of dry extratropical cyclone formation, Beare (2007) found that the spatial maximum of surface stress at every instant scales in a simple way with the initial strength of the jet stream. Furthermore, if the boundary layer parameterization is changed, changes in the minimum pressure correspond closely to changes in the surface stress averaged over the cyclone. Beare (2007) concludes that this demonstrates the important role of the synoptic-scale flow in organising the boundary layer structure. In Boutle et al (2015), this picture
is completed by detailing how barotropic and baroclinic potential vorticity generation act together to reduce cyclone growth;



Boutle et al (2010) add moisture transport and convection to the picture, and show that the spin-down caused by surface friction in a moist cyclone is of similar magnitude to that in a dry cyclone.

### 5.4 The tail of the stress distribution is also regular and its shape can be estimated reliably from climate model simulations

Sections 5.1 and 5.3 imply that the tail of the distribution function of stress is also highly regular; possible saturation of the drag coefficient over sea at high wind speeds (Curcic, 2020; Richter et al, 2021) is not likely to change this; see Section S4. Climate model simulations indicate that this is indeed so: Fig. 5 shows the distributions of annual maxima of stress from 3-hourly RACMOv2.3 and 6-hourly SEAS5 data. Although these two tails clearly differ in scale, they agree in shape, as both are nearly exponential.

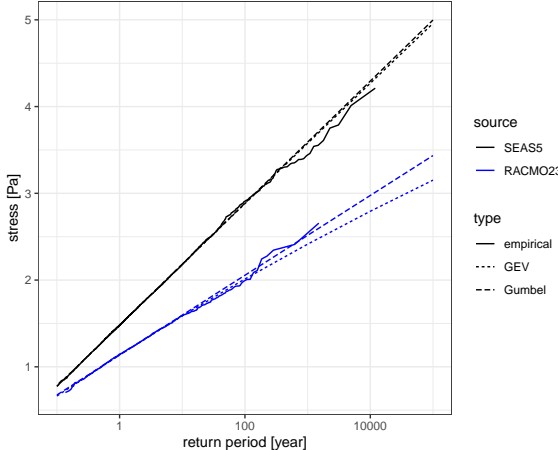

**Figure 5.** Empirical return values of stress from annual maxima at 3.5E, 54N from 3-hourly RACMOv2.3 (blue) and 6-hourly SEAS5 (black) data, with return values from GEV fits (dotted) and from Gumbel fits (dashed).

Both models have different resolutions and output frequencies and the boundary layer parameterisations are quite different, but this appears to have little impact on the shape of the stress tail. This is also evident from Fig. 6, showing estimates of the GEV shape $\gamma$ from annual stress maxima over the North Sea. On average, the estimates from SEAS5 are slightly lower than the estimates from RACMOv2.3, differing on average over the central and southern North Sea by 0.026, which is not much. The close agreement in the tail shape of stress from these models indicates that the shape is determined by other factors than

resolution, output frequency or boundary layer parameterisation, such as jet stream climatology (Section 5.3).

Another relevant check is to compare estimates of the shape of the tail of stress from SEAS5 with those from the reanalyses ERA5 and KNW for the same position 3°E, 55°N in the central North Sea, far from the influence of land; see Section 3. Plotted as functions of sample fraction $p = k/n$, estimates of the GW shape parameter $\beta$ (see (4), (5)) can be regarded as



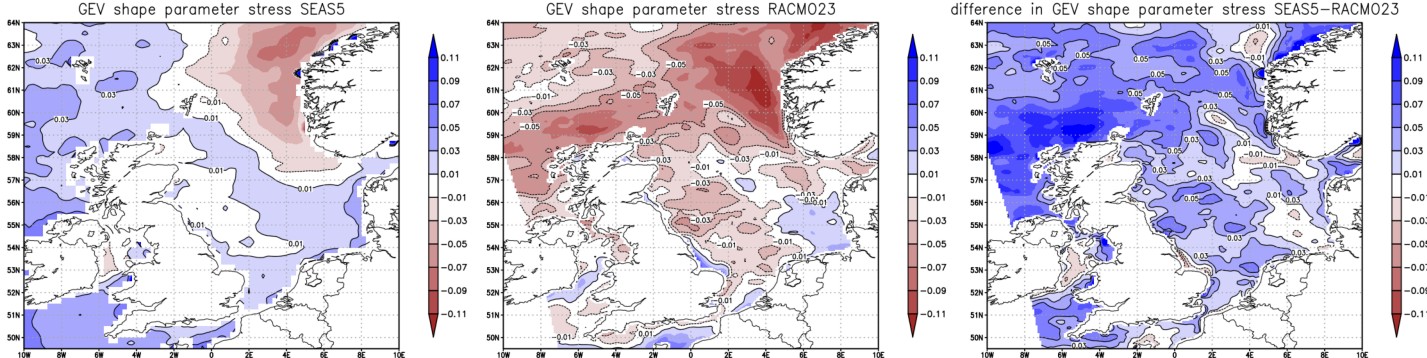

**Figure 6.** Estimates of the tail shape $\gamma$ (see Section 4) from annual stress maxima from SEAS5 (top left) and from RACMOv2.3-downscaling of EC-Earth3bis runs (top right), and their differences (bottom).

estimates of the dimensionless shape function $y\big(\log(yq'(y))\big)'$ with $y = -\log p$. Fig. 7 shows for SEAS5 only a slight increase in these estimates with decreasing $p$ from about 1.2 to 1.35, which indicates a regular tail which is slightly heavier than the exponential tail. Furthermore, the estimates match the much less precise estimates from the two reanalyses which, unlike the SEAS5 reforecasts (with lead times exceeding 1 month), are constrained by weather observations and employ significantly different resolutions and different approaches to boundary layer momentum exchange. This provides further confirmation that current climate models are capable of simulating the shape of the stress tail.

As a consequence, systematic errors in extreme stress from these models should take the form of scale/location errors. This agrees broadly with the analysis in Larsén (2012) of the effect of limited resolution on extreme wind speeds over Danish and German coastal waters from mesoscale models; for a Gaussian process as approximation of wind fluctuations, they argue that the correction should have the form of a scale adjustment. Although Larsén (2012) do not consider the effects of differences in boundary layer parameterization of the mesoscale models used for downscaling of global low-resolution analyses, their corrected estimates of mean annual maximum wind speed from these models are mutually compatible and in line with wind measurements. In climate model simulations and in SEAS5 (for forecast ranges exceeding 1 month), storms are generated by the model without assimilation of measurements, which offers much additional freedom to deviate from observed climatology. Our findings indicate that even in this case, the shape of the stress tail generated by such models is reliable over a wide range of return periods.

# 6 How reliable are the shapes of the tails of HW along the Dutch coast derived from simulated data?

Since stress provides the main forcing of a storm surge, it can be expected that the tails of the surge and HW along the Dutch coast are also regular. They should in fact be similar to the tail of stress, as storm surge is roughly proportional to the stress if we ignore the effects of wind directionality, fetch and duration limitations and along-shore flow. Indeed, empirical tails (e.g.



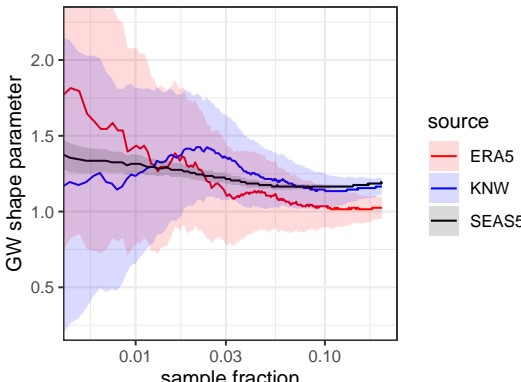

**Figure 7.** Estimates of the GW shape parameter $\beta$ (see Section 4) of stress from SEAS5 with 95% confidence intervals (black) and from the KNW reanalysis (blue) and the ERA5 reanalysis (red), along with their confidence intervals at $3°$E, $55°$N (central North Sea). The horizontal axis indicates the sample fraction $k/n$ used for the estimation (see text).

Fig. 1) do not suggest anomalies. Whether their shapes can be reliably estimated from hydrodynamic model simulations forced
by model-generated stress depends on the nature of bias in these models. Validation of a similar, higher resolution version of
DCSM in Zijl et al (2022); Zijl and Laan (2021) shows that predicted skew surge at tidal stations on the Waddenzee (bounded
by the labels 4-7 in Fig. 2) has a negative bias for surge above a threshold close to the 99% quantile; the magnitude of the bias
increases approximately linearly in the excess of surge above this threshold; see also Section S8. Model resolution does not
appear to have much systematic impact on the bias for most the coastline (van den Brink, 2020); in fact, a similar bias pattern
was already reported in Ridder et al (2018). This suggests either an issue with the representation of hydrodynamic processes
in the shallow Waddenzee estuary under severe storm conditions (e.g. wave-current interaction, changes of the seabed and
its roughness) or insufficient resolution of mesoscale variations of stress, which would affect the Waddenzee more than the
adjacent North Sea. Modelling experiments with varying forcing and resolution point to the former explanation (van den
Brink, 2020), but the issue has not been resolved yet.
However, the apparently linear change of the bias in skew surge above a threshold suggests that the shapes of surge and
HW tails estimated from the model simulations may still be accurate above the threshold. To check this, GW tail shape
estimates from tide gauge data and simulations are compared in Fig. 8. Indeed, the estimates from the SEAS5/DCSMv5 data
are compatible with the estimates from tide gauge data for 5 of the 6 tide gauge stations. The exception is Delfzijl (6) on the
small Eems-Dollard estuary (Fig. 2). The shape estimates from the measurements from this station and from Harlingen (3)
are highly irregular at sample fractions below 0.01, which is not plausible in view of the regularity of the estimates at other
stations nearby and the considerable spatial coherence in storm surge on the North Sea. The wide confidence intervals of these
estimates indicate that they are not reliable. This suggests that the use of SEAS5/DCSMv5 data for shape estimation may help
to ensure that spatial variations in tail shape are plausible. The poor match at high sample fractions above say 0.01 for Delfzijl



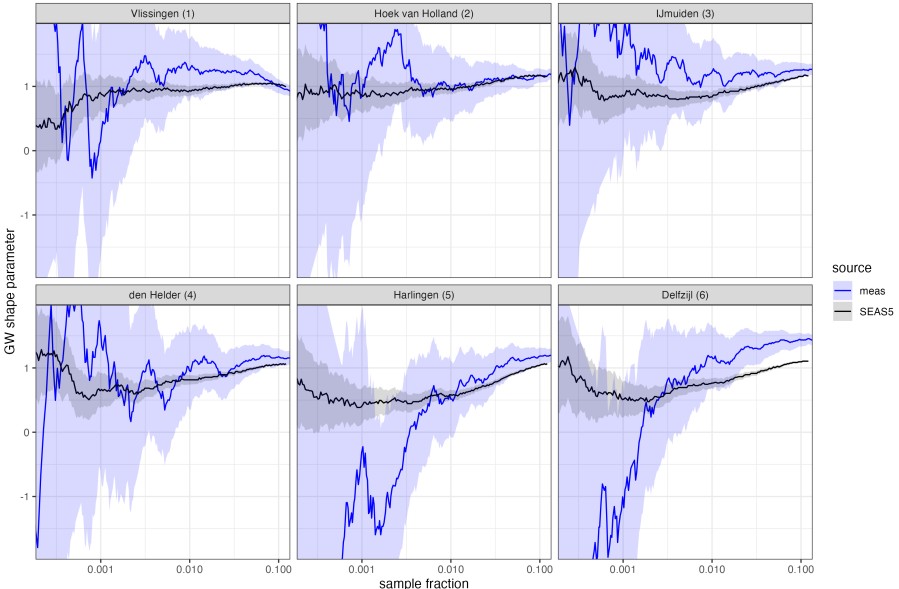

**Figure 8.** Estimates of the GW shape parameter $\beta$ (see Section 4) of HW vs. sample fraction with 95% confidence interval from measurements (blue) and from SEAS5/DCSMv5 simulations (black) for 6 tide gauge stations (see Fig. 2).

may be related to the change in bias in model predictions of skew surge from negligible to negative at roughly the 99% quantile found in Zijl et al (2022); Zijl and Laan (2021) (Section S8), which will distort shape estimates at mentioned sample fractions.

## 7 Application to the extreme value analysis of stress and coastal water level

### 7.1 Stress

Based on the findings of section 5, return values of stress at the locations marked by triangles in Fig. 2 were estimated from SEAS5 stress. First, the tail of the distribution function of stress was estimated from all data available for the location by fitting a GW tail (5) (see Section 4). Based on simulations in de Valk and van den Brink (2023a), the estimates from a sample fraction of 0.012 were selected; see also Section S7.1. Subsequently, probabilities (fractions of time) of exceedance $p$ were converted to frequencies $\mu$ and return periods $T = 1/\mu$ by $\mu = p\alpha/\Delta$ with $\Delta = 1/(24 \times 365.25)$ the sampling interval of hourly wind measurements in years and $\alpha$ their extremal index, to be interpreted as the reciprocal of the mean size of a cluster of high values; see Leadbetter et al (1983). We chose $\alpha = 1/2$. The motivation for this computation of $\mu$ is the following: (a) the fraction of time $p$ has the same meaning regardless of sampling interval (this is true regardless of systematic differences between estimates from data from different sources, which should be dealt with separately); (b) estimates suggest that hourly wind measurements have an extremal index $\alpha$ of 1/2 to 1 (see Section S6); (c) for stress, the extremal index should be the same as for wind speed; (d) taking into account that nearshore wave conditions vary on longer time-scales than the stress, we adopted the lower value



1/2. There is reason to believe that wind is not very different from a Gaussian process (Larsén, 2012). Just as for a Gaussian
process, there is therefore reason to question whether the concept of a fixed extremal index is appropriate for wind speed and
stress (Leadbetter et al, 1983). However, (d) above still makes practical sense.

Return values were also estimated for stress magnitude restricted to wind direction bins of $22.5°$ (see Section S7.2). Finally,
all return values for stress were increased by the same factor of 1.1, based on an earlier analysis of the effects of atmospheric
model resolution on skew surge in van den Brink (2020).

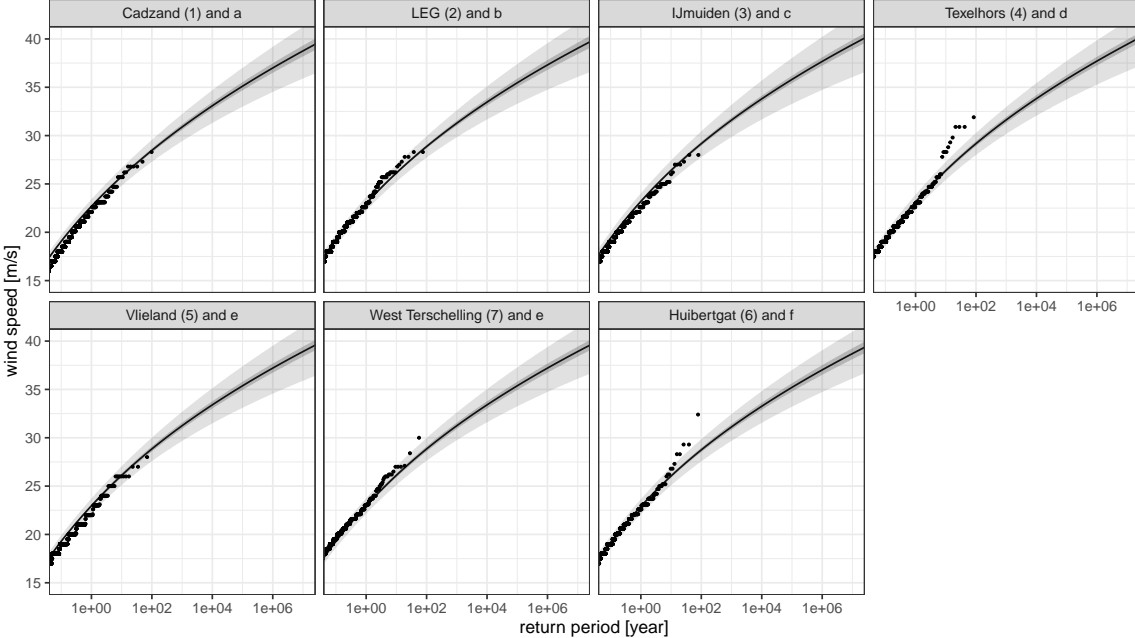

**Figure 9.** Empirical return values of wind speed $u_{10}$ restricted to wind directions in W to N from measurements at seven stations (dots; see
headers), and corresponding provisional estimates derived from the corrected SEAS5 stress by the Charnock (0.025) relation, with sampling
error (95% confidence intervals; dark grey) and assessment of total uncertainty (idem, light grey). Numbers and letters in the headers refer
to the sites marked by squares and triangles in Fig. 2.

The estimates of the tails of stress (see Section S5) cannot be validated directly. As a sanity check, we compared the
tail distributions of wind speed $u_{10}$ obtained from stress using an approximation formula to empirical distributions from
measurement data. For the approximation, we used a logarithmic wind profile with roughness length $z_0$ determined from the
Charnock relationship with a constant of 0.025. The return values of wind speed associated with wind directions in the sector
from W to N (which are not disturbed much by land) were compared with the empirical return values determined from hourly
wind observations at nearby measurement stations; see Fig. 9. Here, the return period of the $k$-th highest observation (circle)
is determined as $L/(k\alpha)$ with $L$ the record length in years and $\alpha = 1/2$ as above (this makes the comparison independent of
the choice of the extremal index).





With the chosen Charnock constant, the overall agreement is surprisingly good. Had we chosen a lower Charnock constant of 0.0185 as in Wu (1982), which is the default of the SWAN wave model used to derive nearshore wave conditions (see

Section 2) if forced by wind, we would have arrived at return values that markedly exceed the empirical values. That we need a Charnock constant higher than the Wu (1982) value to approximate the empirical tails of wind speed well (including those of the offshore station LEG) indicates that the correction factor of 1.1 applied to the stress is not too low for our purpose. The return values for long return periods in Fig. 9 are of course not reliable, as they depend sensitively on the assumed Charnock relation; these are mainly shown to indicate the uncertainties in terms of wind speed (see Section 7.3).

Since these wind speed tails vary little along the coast (Fig. 9) while consistent with measurement data, it seems likely that the large spatial variation in return values found in earlier analyses of measurements (Caires, 2009; Wieringa and Rijkoord, 1983) is largely due to sampling variability.

To make validation easier, we could have opted to derive statistics of extreme wind speed instead of stress from the SEAS5 wind data, and force the SWAN nearshore wave model (Section 2) with wind. However, the near-surface wind speed $u_{10}$ from

numerical models is only a diagnostic variable derived using one particular drag formulation; if this wind would be used in the SWAN wave model, SWAN would convert it back to stress by a different (and much simpler) drag formulation than used in SEAS5. Our choice of estimating return values of stress instead of wind speed avoids these ambiguities and inconsistencies. Furthermore, tails of $u_{10}$ from models certainly do not show the universality of shape exhibited by tails of stress (see Section 5.4).

**7.2   High water**

For HW at the six tide gauge stations (see Fig. 2), almost the same method was used as for stress, with one important difference: based on the conclusion of Section 6, the shape parameter of the tail of HW was estimated from SEAS5/DCSMv5 data, with the WAQUA DCSMv5 model driven by the original, uncorrected shear stress from SEAS5 (see Fig. 8). With this shape fixed, the GW tail (i.e., the scale and location parameters) was subsequently estimated from the tide gauge data. Just as for stress, a

relatively large sample fraction of 0.012 was used, which for HW gives relatively high values of the shape parameter (see Fig. 8). The extremal index of the HW observations was estimated to be 1, so clustering of consecutive HWs effectively vanishes for increasingly rare events; see Section S6.

The estimated return values of HW are shown in Fig. 10 together with the empirical (trend-corrected) return values. The high values reached in Vlissingen (1), Hoek van Holland (2) and IJmuiden (3) reached during the severe 1953 flood stand out.

For Hoek van Holland, this value has an estimated frequency of exceedance $\mu$ of 0.0011 /yr, so the probability that an annual maximum exceeds it in 132 year (the record length) is $1 - \exp(-132\mu) = 0.14$. Therefore, the 1953 water level is definitely no outlier. Earlier estimates of the frequencies of exceedance from only measurements as in Dillingh et al (1993) were higher, as the shape estimates were influenced by the high HW observed during the 1953 storm. By estimating the shape from simulated data, the present analysis avoids an outsized influence of single events.

As a further check, we computed for every station the ratio of the scale parameter estimated from tide gauge measurements and the scale parameter estimated from SEAS5/DCSMv5 data; see Tab. 1. If bias in the tail of HW from SEAS5/DCSMv5 is



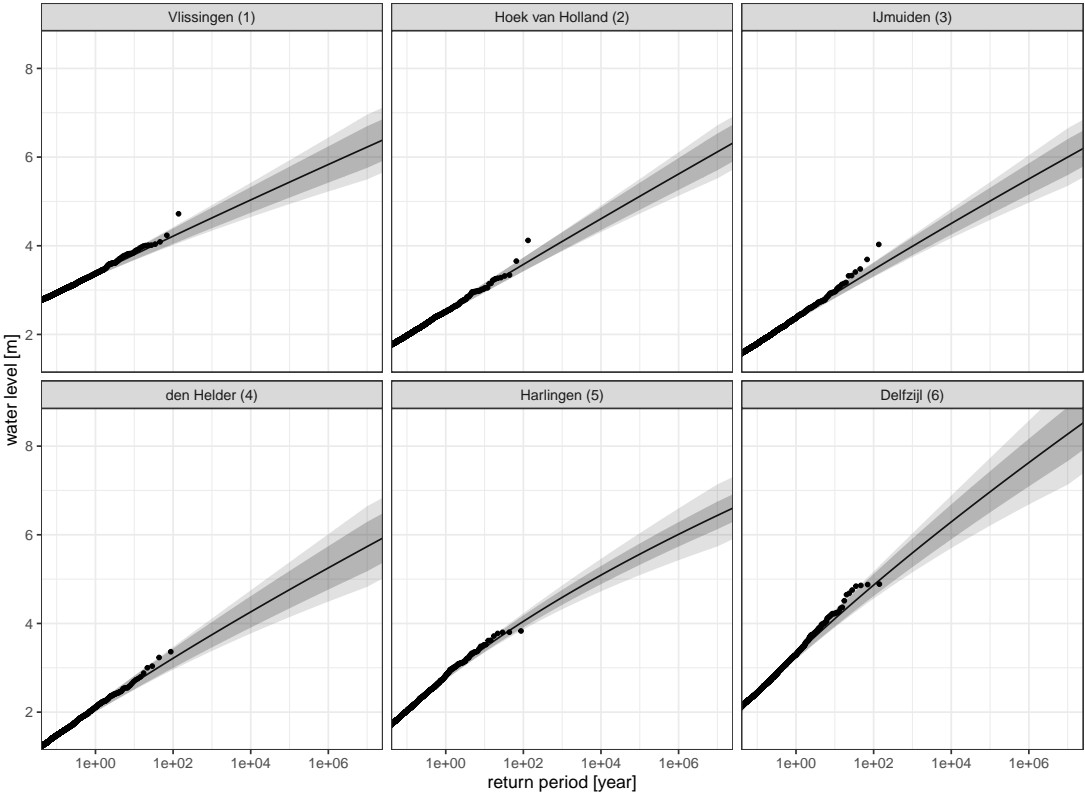

**Figure 10.** Estimated return values of water level at six tide gauge stations (see Fig. 2) with with sampling error (95% confidence intervals; dark grey) and assessment of total uncertainty (idem, light grey), with empirical return values from tide gauge data (dots).

only due to bias in the SEAS5 stress, then this ratio indicates the factor by which the SEAS5 stress should be corrected. The outlying value for Vlissingen can be explained by spatial mismatch: the DCSMv5 model has only a few output points in the small Westerschelde estuary. The scale ratios for Hoek van Holland and IJmuiden along the West coast are close to the stress

correction factor of 1.1 arrived at earlier based on other arguments, indicating that the ratios for these sites are to a large extent explained by bias in the uncorrected SEAS5/DCSMv5 stress. In fact, running the DCSMv5 model with SEAS5 stress fields inflated by a factor of 1.1 and repeating the analysis gives scale ratios which are 0.09 lower, which shows that the increase of the scale parameter of the tail of HW from DCSMv5 is almost proportional to the stress increase. Possibly, a stress correction by a factor of 1.15 would have been slightly better than the applied factor of 1.1.

For Den Helder (4), Harlingen (5) and Delfzijl (6) on the Waddenzee estuary (Fig. 2), larger ratios are found. This could be due to errors in the stress forcing of the DCSMv5 model in this area (e.g. due to the limited resolution of SEAS5), or to bias in DCSMv5 when applied to this shallow estuary with complex bathymetry. A study of the effects of dynamic downscaling of the atmospheric forcing and of hydrodynamic model resolution (van den Brink, 2020) finds no large systematic impact of resolution (see also Section 6). The current working hypothesis is therefore that under severe storm conditions, certain



process(es) in the shallow Waddenzee are not adequately represented in current shallow-water flow models; investigating this
further is beyond the present scope.

| tidal station | SEAS5 stress | inflated stress |
|---|---|---|
| Vlissingen (1) | 0.91 | 0.85 |
| Hoek van Holland (2) | 1.14 | 1.05 |
| IJmuiden (3) | 1.15 | 1.06 |
| Den Helder (4) | 1.24 | 1.15 |
| Harlingen (5) | 1.22 | 1.14 |
| Delfzijl (6) | 1.28 | 1.19 |

**Table 1.** Scale ratios (rations of scale parameters of the tails of HW estimated with and without using measurement data for location and
scale), for HW data simulated from SEAS5 stress, and simulated from SEAS5 stress inflated by a factor of 1.1.

For the dependence of the tail of HW on wind direction, we made estimates from SEAS5/DCSMv5 data only, and from
a combination of SEAS5/DCSMv5 data (for shape) and HW and wind direction measurements (for scale/location). Both are
shown in Fig. 11 for two stations. The differences are small. For wind from land, the estimates from only SEAS5/DCSMv5
data are lower, indicating a systematic overestimation of stress from these directions. This is most likely an effect the limited
resolution of SEAS5.

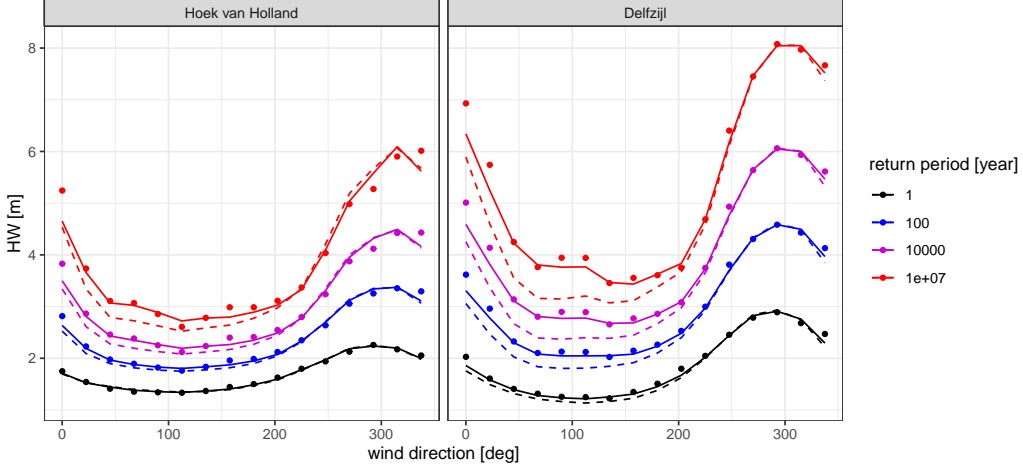

**Figure 11.** Estimates of return values for HW coincident with wind direction in a $22.5°$ bin for return periods of 1, 100, 10,000, and $10^7$
years for Hoek van Holland (left) and Delfzijl (right) with the directional dependence estimated from SEAS5/DCSMv5 only (dashed), using
scale/location estimates from measurement data (dots) and the latter after additional smoothing (full).



### 7.3 Uncertainty of the estimates

The sampling error of the estimated return values for return periods of $10, \ldots, 10^7$ years has been assessed using a bootstrap method (Litvinova and Silvapulle, 2018, 2020; de Haan and Zhou, 2022). More specifically, we use the block bootstrap (Kün-
395 sch, 1989) to estimate variances, which is applicable in the case of serially dependent time series; see Section S7.3 for further details. Confidence intervals are based on the normal approximation.

Simultaneous values of stress from different SEAS5 ensemble members which contain a high value(s) are effectively independent (see Section S2), so ensemble members were treated as independent in the bootstrapping (see Section S7.3). Since all 25-50 ensemble members initiated in the same month of the same year are based on the same analysis, there is some residual
dependence between forecasts for the same year/month in the form of a common modulation of storminess, which is not accounted for in the bootstrap procedure. This was found to have a minor effect on the precision of the tail estimates (de Valk and van den Brink, 2023b). The confidence intervals representing the sampling errors in return values are shown in Fig. S6 (Section S5) for stress, in Fig. 9 for the speed of W to N wind $u_{10}$ provisionally derived from stress, and in Fig. 10 for HW (dark grey). For stress and derived wind, the confidence intervals are very narrow, but not for HW, as its scale/location are estimated from
tide gauge measurements.

In addition, the error analysis addresses model-related uncertainty: unknown systematic errors in the estimated tail distributions of wind shear stress and HW from simulated data resulting from limitations of the models such as resolution and parameterisations of processes. The model-related uncertainty in the GW tail distribution is represented as a normally distributed error in the shape parameter. The reason for this choice is that errors in estimates of return values for long return
periods are primarily determined by errors in the shape parameter (Section 4). Furthermore, for HW, model-related errors in scale/location are corrected using measurement data.

For shear stress from SEAS5, we assume that the shape parameter of the omnidirectional GW tail is normally distributed with mean 0 and standard deviation 0.1 (likely within $\pm 0.1$, very likely within $\pm 0.2$). This value is a crude estimate based on a comparison of estimates of the shape parameter $\gamma$ (see Section 4) from annual maxima of SEAS5 stress and of the
415 RACMOv2.3-downscaling of EC-Earth3 runs for the same area around the Netherlands (see Section 3), which differ considerably in resolution and drag relation; see Fig. 6. The mean difference over the southern and central North Sea is 0.026, from which we estimate the standard deviation as $\sqrt{0.013^2 \times 2} = 0.018$. Using equation (4), this results in 0.12 (rounded to 0.1) for the standard deviation of the GW shape parameter. For HW, we employ the simplification that HW is roughly an affine function of stress (for small sample fractions, both in the hydrodynamic model and in reality; see Section 6), so the errors in the GW
shape parameter of stress and HW are roughly equal. The model-related uncertainty is included as a random disturbance to the shape parameter estimates in the bootstrap procedure.

Confidence intervals of the resulting total error are shown in Fig. S6 (Section S5) for stress, in Fig. 9 for the speed of W to N wind $u_{10}$ provisionally derived from stress, and in Fig. 10 for HW (light grey). In particular for stress and derived wind, the model-related error broadens the intervals considerably. However, the confidence intervals for HW at Hoek van Holland
are much narrower (by a factor of about 4) than for estimates from measurements alone; e.g. compare with those in Fig. S1



in Section S1 and in Fig. 1 for estimates based on measurements only. Comparing the latter two, we see that fitting a GW tail instead of a GP tail (Section 4) on the measurement data already reduces the uncertainty considerably, but a much larger further reduction is achieved by using shape estimates from SEAS5 data, even when accounting for the model-related error. This is qualitatively in line with earlier simulation results in de Valk and van den Brink (2023a).

### 7.4   Local dependence between simultaneous stress and coastal water level extremes

The SEAS5/DCSMv5 data can also be used for estimating the strength of the dependence between HW and (nearly simultaneous) stress or wind speed. Different models have been proposed for this dependence. Here, we discuss a particularly simple class of models within the framework of Ledford and Tawn (1996, 1997, 1998); Wadsworth and Tawn (2013); de Valk (2016b), which can distinguish weaker forms of dependence than classical extremal dependence (e.g. Rootzén and Tajvidi , 2006). For two random variables $X_1$ and $X_2$ with distribution functions $F_1$ and $F_2$, respectively, it implies that the tail of the distribution function of $-\log\max(1-F_1(X_1), 1-F_2(X_2))$ approximates an exponential distribution with scale parameter $\eta \leq 1$ (de Valk, 2016b, eq. (4.4)). To help the interpretation, we discuss $\rho := 2\eta - 1$, which for a bivariate normal distribution happens to agree with the ordinary Pearson correlation coefficient (but is estimated in a completely different manner).

Fig. 12 (right) compares values of $\rho$ derived from estimates of $\eta$ for HW and wind speed or stress restricted to bins of wind direction from measurement data of wind $u_{10}$ and from SEAS5/DCSMv5 data of stress. The estimates are nearly equal, so the SEAS5 data appear to be suitable for estimation of the extremal dependence. In fact (not shown) it makes very little difference whether data of wind speed or stress from SEAS5/DCSMv5 are used for this purpose. The estimated variation of $\rho$ with wind direction makes sense: strongly positive for wind from sea, and negative for wind from land. Furthermore (left), the estimates from SEAS5/DCSMv5 data are insensitive to the sample fraction used for the estimation of $\rho$. This indicates that the dependence model class considered here provides a good representation of the extremal dependence between wind speed and high HW. Using measurement data, this type of check would be much less effective, as measurement records are too short for this purpose.

## 8   Conclusions and outlook

Even a measurement record of about 140 years (considered long in meteorology) is not enough to reasonably constrain return values for long return periods of 10,000 years or higher (Section 2). This case study demonstrates that the uncertainty in the statistical modelling of extreme midlatitude shear stress and coastal water level can be reduced considerably (for water level at Hoek van Holland, by about a factor of 4) by the cautious use of large archives of weather data simulated by state-of-the-art models (Sections 5-7).

This error assessment includes model-related error in return value estimates (Section 7.3), which is inherently difficult to quantify; hence our assessment is crude. However, had we used only measurements for the estimation of return values, then the uncertainty would be much larger and also difficult to assess reliably, as demonstrated by the example in Section 2.



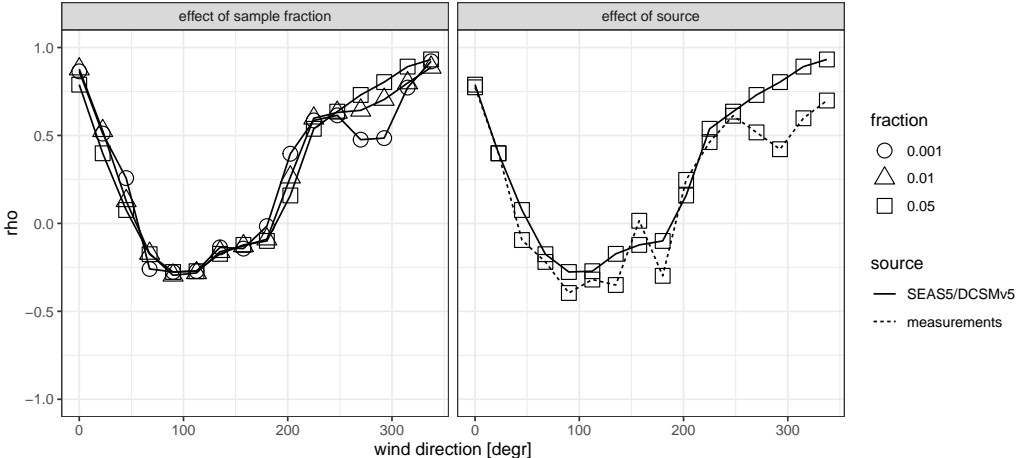

**Figure 12.** Estimates of the strength of extremal dependence $\rho$ (see text) between simultaneous wind speed or stress and HW restricted to wind direction in $22.5°$ bins. Left: estimates from SEAS5/DCSMv5 data of stress and HW from point a in Fig. 2. Right: same estimates from SEAS5/DCSMv5 data, and estimates from measurements of wind and HW from Hoek van Holland/LEG (2 in Fig. 2).

The uncertainty reduction is achieved by estimating the tail shapes from about 8000 years of SEAS5 weather re-forecast data and water levels simulated from these data. We substantiate that these tail shape estimates are reliable (Sections 5-6); for stress, they show very little dependence on model resolution and drag formulation, and shape estimates are generally compatible with

estimates from measurements or reanalysis data (Sections 5-6).

Scale and location of the tails of HW at the tide gauge stations can be accurately determined from measurement data. For other sites along the coast, corrections of the location and scale estimates need to be derived from the corrections for the tide gauge stations by interpolation.

For stress, scale and/or location bias is a bigger issue, because there are no stress measurements for correction and valida-

tion. Currently, stress is corrected by a factor of 1.1, in reasonable agreement with the HW scale parameter corrections for tidal stations on the West coast in Tab. 1. For the purpose of transforming return value estimates of stress to nearshore wave conditions using the SWAN model (Section 2), this factor appears to be sufficient (see Section 7).

Substantial local bias in the scale parameter of HW for stations along the Waddenzee estimated from only SEAS5/DCSMv5 simulations indicates limitations in modelling of storm surge on this shallow estuary which are presently not well understood

(see Section 7); more research is needed to identify and correct the source(s) of error. At present, the estimation of scale and location from measurement data provides a solution.

The use of very large simulated datasets in this study produces spatially consistent return value estimates which are insensitive to individual events such as the 1953 storm, without having imposed spatial smoothness.



Dependencies between variables in the extreme ranges of stress and/or HW (dependence on wind direction and between stress and HW) can be reliably estimated from simulated data (Section 7).

The present method, developed for the Dutch coast, may be applicable to other coastlines along the North Sea and possibly to other regions with similar climates. However, careful review and checking of the assumptions and their consequences for the application is essential. A potential advantage of very large archives of simulated weather is good coverage of the phase space, but this will only materialise if all relevant phenomena (e.g. storm types) are faithfully represented in the data.

In the future, further model improvements may lessen the need for scale/location adjustments relying on measurement data and make it easier to produce reliable return level estimates at all points along the complex coastlines with high resolution.

For the protection of the Netherlands against flooding, extreme hydraulic loads on dikes along the rivers and lakes are also an important issue. It is a major challenge to obtain reliable estimates of stress over these smaller water bodies (Sterl, 2019b). Furthermore, dynamic downscaling of 8000 years of SEAS5 data to resolve the effects of land-water boundaries on the stress would require excessive computing resources. Possibly, machine learning might help to reduce the effort (Wang et al, 2021; Doury et al, 2023). An additional benefit could be that high-resolution stress and wind are also produced over land, extending the options for checking and correction of bias in the severity of simulated storms by making use of inland wind measurements. Furthermore, it may be possible to extend the current approach to precipitation within the large Rhine and Meuse catchments, in order to improve the estimation of return values of discharge of these rivers as well as possible associations between extreme discharge and other variables like stress and coastal HW.

*Code availability.* The R code for estimation of GW and GP tails is available at https://github.com/ceesfdevalk/EVTools; GW tail estimates are computed with FitGW_iHilli.R.

*Author contributions.* Conceptualization, investigation, writing: CdV and HvdB; methodology: CdV.

*Competing interests.* The authors declare that they have no conflict of interest.

*Acknowledgements.* We like to thank Marcel Bottema and Robert Slomp (Rijkswaterstaat), Pieter van Gelder (TU Delft) and Andreas Sterl (KNMI) for their helpful comments and suggestions in the course of this study.



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
