# Peer review of "An appraisal of the value of simulated weather data for quantifying coastal flood hazard in the Netherlands"

_EGUsphere, 2024_

## Author Response (AR1)

**Response to reviewers**

Cees de Valk, Henk van den Bribnk

January 31, 2025

**1    Reply to RC1**

We are grateful for the helpful comments of the referee on the structure and wording of the manuscript, and have tried to implement the suggested changes. In particular,

(a) We have restructured and simplified the introduction to focus directly on the central question of whether simulated weather data can be used to improve the accuracy of return value estimates, and then discuss the specific case to be considered (coastal water level, wind stress, Dutch coast, etc.). Technical concepts which need definition (e.g. "tail") are removed from the introduction. A basic introduction of some concepts (e.g. the meaning of a probability distribution in relation to frequencies of exceedance of levels) is added in Section 4.1.

(b) We removed the discussion of directional dependence and of dependence between stress and high water (HW) from the manuscript.

(c) In Section 4, the methods used to address the problem (estimators, uncertainly estimation by bootstrapping) are introduced.

(d) We still feel that discussing both water level and stress is useful: the joint analysis of stress and HW adds substantial value to the analysis of each variable individually. Without considering HW, the practical relevance of the manuscript to coastal flood risk (and hence, to the journal) would be limited. Also, HW is used to check the estimates of the scale correction applied to the tail of stress, which is important because there is no alternative for it. On the other hand, without considering stress, the connection between extremes of HW and the generating storms would be missing, and therefore an essential part of our argument that the shape of the tail of HW can be determined from simulated data would be missing.

(e) We have tried to simplify and structure the presentation of these different aspects, for example by including a diagram summarising the proposed method (containing the various datasets and processing involved).

**2   Reply to RC2**

We are grateful to the referee for the helpful comments and suggestions for improvement of the structure and writing. In response, we have made the following changes:

(a) We have restructured and simplified the introduction to focus directly on the central question of whether simulated weather data can be used to improve the accuracy of return value estimates, and then discuss the specific case to be considered (coastal water level, wind stress, Dutch coast, etc.). Technical concepts which need definition (e.g. "tail") are removed from the introduction. A basic introduction of some concepts (e.g. the meaning of a probability distribution in relation to frequencies of exceedance of levels) is added in Section 4.1.

(b) References to later sections have been removed.

(c) In Section 4, the methods used to address the problem (estimators, uncertainly estimation by bootstrapping) are introduced.

(d) We have added a diagram summarising the proposed method and the data and processing involved.

(e) Following the suggestion of another reviewer, we dropped the discussion of directional dependence and of dependence between stress and high water (HW) from the manuscript in order to improve focus.

(f) We checked and improve the writing and phrasing and corrected the citations.

**3   Reply to AC1**

We are grateful to dr. Agustín Sánchez-Arcilla for contributing valuable comments on our manuscript. We checked and improved the structure and wording and corrected the references. Below, we will answer the detailed comments point by point, citing the comment.

1. Whether the shape of the probability distribution and the characteristics of the tail will be the same under future climates, particularly in the case of disruptions.

   A: This is an important issue, but would require an entire study in itself. One could analyse very large datasets of climate model projections for the present and future epochs, and compare shape estimates from both. This would also need to involve several models to check if results from different models are consistent. We would recommend this as a follow-up to the present study: the current manuscript is already quite a long read, and to us it makes sense to study the use of simulated data to estimate return values for present climate (as we do in our manuscript) before moving to the topic of possible effects of climate change, which could involve greater

prominence of cyclones of (sub)tropical origin (If remnants of tropical cyclones are not considered, we hypothesize that the shape will not change (much) under climate change, as the physical principles will be unchanged for the storms of extratropical origin).

2. The analysis of tail shape differences using the Gamma parameter or the Beta parameter should be better explained, not only making reference to the data source or probability distribution but also to the underlying physics.

   A: Explaining the tail shape of stress based on physical principles is something we would very much like to do. However, we have not been able to do this until now. The cited literature indicates that the shape of the tail of stress is primarily controlled by the statistics of the jet stream velocity and position, even as other processes may strengthen or weaken the cyclone over its lifetime.

3. The assumptions that the development of extra tropical cyclones will constitute a continuum under future climates and that tail anomalies will not be affected by different extreme populations, should also be discussed making reference to future climate projections, particularly depending on scenario and horizon.

   A: Indeed, the literature indicates that storms of (sub)tropical origin may become more prominent in the future, which may require explicit consideration of different populations of cyclones. However, we do not intend to discuss future climates at this moment, as it will widen the scope of the manuscript too much and will require much additional research. It will certainly be a focus of future work, as the problem is highly relevant. We have a added a few lines about this under "Conclusions and outlook".

4. The discussion about the benefits of increasing resolution should also make reference to the level of numerical diffusivity that is associated to resolution. This is an important point for all the following discussion on the shear stress which is directly related to the energetic transfer between air and sea.

   A: Numerical diffusivity is indeed an important aspect of resolution, depending also on the discretisation scheme applied: due to numerical diffusion, the effective resolution is often coarser than the mesh size. We consider this background knowledge that does not warrant a discussion in our manuscript. For the most important layers close to the sea surface, vertical momentum transfer has been parameterised in numerical weather prediction models, so numerical diffusion will not play a role here. This is the reason that we focus on differences between drag parameterisations employed in the models discussed. Additionally, the wind fields that generate the extreme surges are much larger than the model resolution, which reduces the influence of horizontal numerical diffusion on the outcomes.

5. The difference in stress tail shape depending on resolution (e.g. figure 5 and associated paragraph) should also be discussed in more detail, explaining the large difference in scale and relating that to the underlying physics.

A: We agree with this comment. Indeed, we cannot relate the difference in scale to resolution: based on resolution only, we would expect the difference in scale to have the opposite sign of what we observe. Therefore, the difference in scale is much more likely related to the different drag parameterizations in these models, which are indeed very different. We added this point to the revised manuscript.

6. The assumption that the extremal index for stress should be the same as for wind speed (line 320 and following) should also be discussed considering the non linear relationship and the reasons to question a single extremal index for both variables.

A: Nonlinearity of the relationship between u10 and stress does not affect the extremal index, and it does not affect estimators of the extremal index: they are invariant to a continuously increasing transformation. Therefore, we do not intend to elaborate on this point.

7. The conclusion that large spatial variations in return values, from measurement analysis, is due to sampling variability is rather questionable (line 345 and following). This should be discussed considering sources of spatial variability and the dynamic evolution of the storm event over different spatial positions.

A: We consider sampling variability a likely explanation of the large spatial variability of return value estimates from measurements in Caires (2009) because (a) the samples are relatively small (typically covering about 40 years), and in addition (b) the thresholds for these estimates were selected by an adaptive scheme, which selects a threshold based on the observed variation of estimates as functions of threshold. Furthermore, the largest "apparent outliers" in Figure 9 are not very large and each appears to be limited to a single site; at neighbouring sites, the simultaneous values tend to be much lower. Furthermore, several "outliers" at a particular site are related to the same storms. Therefore, we suspect that they are the result of sampling variability and possibly in some cases erroneous and/or related to specific local conditions at the site (e.g. funneling; most sites are at or near the coast). In response to the above comment, we intend to address this issue in slightly more detail in the revision of the manuscript.

8. The same discussion should be carried out for the limited universality of U10 tail shapes when compared to stress tail shapes. A physical based discussion on how the variability from the wind speed is transferred to the stress field, considering their non linear relation would greatly benefit that paper.

A: The limited universality of u10 tail shapes is a topic we discussed in an earlier report (de Valk and van den Brink, 2023b) in some more detail. We have removed it

from the manuscript: demonstrating it would require additional figures and discussion of wind speed data from model simulations and their tails, which would expand the already lengthy manuscript further. We find this topic interesting but not essential.

9. The scale ratios from a number of positions along the coast (line 375 and following) should also be discussed considering the specific geographical constraints of each of those locations, particularly when in some of the locations larger ratios are found (line 380). I think that finding there is no large systematic impact of resolution is probably too strong a conclusion since very little is said about the conditions applied for that assessment.

   A: For this reason, we do not explicitly exclude a resolution effect, and only present the alternative explanation of inadequate representation of the hydrodynamics during severe storm conditions in the shallow Waddenzee as a working hypothesis. Furthermore, we now have more detailed figures showing that the effect of hydrodynamic model resolution on bias in surge and water levels is linear, and therefore resolution of the hydrodynamic model does not offer an explanation. These have been added to Section 6.

10. The systematic overestimation of stress from land sectors appears to be attributed to limited resolution of SEAS5 (line 390 and following). I wonder if that is the main reason since land roughness could also play a role in here.

   A: Following the advice of another reviewer, the topic of directionality has been dropped entirely from the manuscript. Indeed, errors in roughness over land may contribute, but the difference in roughness over land and sea is very large: the roughness length over sea is about 100 times smaller than over land; whereas variation over land is of $O(10)$. Therefore, we expect that the poor approximation of the surface roughness due to limited resolution of SEAS5 is the main cause (the model uses a weighted average of the roughnesses over land and sea within a grid cell).